# Individual and Co-Expression Patterns of FAM83H and SCRIB at Diagnosis Are Associated with the Survival of Colorectal Carcinoma Patients

**DOI:** 10.3390/diagnostics12071579

**Published:** 2022-06-29

**Authors:** Tae Young Jeong, Hae In Lee, Min Su Park, Min Young Seo, Kyu Yun Jang

**Affiliations:** 1Department of Medicine, Jeonbuk National University Medical School, Jeonju 54896, Korea; jlhsmo@naver.com (T.Y.J.); krhaein99@naver.com (H.I.L.); pamisu1011@naver.com (M.S.P.); smy5644@naver.com (M.Y.S.); 2Department of Pathology, Jeonbuk National University Medical School, Jeonju 54896, Korea; 3Research Institute of Clinical Medicine of Jeonbuk National University-Biomedical Research Institute of Jeonbuk National University Hospital, Jeonju 54896, Korea

**Keywords:** colorectum, carcinoma, immunohistochemistry, FAM83H, SCRIB, prognosis

## Abstract

Background: FAM83H is important in teeth development; however, an increasing number of reports have indicated a role for it in human cancers. FAM83H is involved in cancer progression in association with various oncogenic molecules, including SCRIB. In the analysis of the public database, there was a significant association between FAM83H and SCRIB in colorectal carcinomas. However, studies evaluating the association of FAM83H and SCRIB in colorectal carcinoma have been limited. Methods: The clinicopathological significance of the immunohistochemical expression of FAM83H and SCRIB was evaluated in 222 colorectal carcinomas. Results: The expressions of FAM83H and SCRIB were significantly associated in colorectal carcinoma tissue. In univariate analysis, the nuclear expressions of FAM83H and SCRIB and the cytoplasmic expression of SCRIB were significantly associated with shorter survival of colorectal carcinomas. The nuclear expressions of FAM83H and SCRIB and the cytoplasmic expression of SCRIB were independent indicators of shorter cancer-specific survival in multivariate analysis. A co-expression pattern of nuclear FAM83H and cytoplasmic SCRIB predicted shorter cancer-specific survival (*p* < 0.001) and relapse-free survival (*p* = 0.032) in multivariate analysis. Conclusions: This study suggests that FAM83H and SCRIB might be used as prognostic markers of colorectal carcinomas and as potential therapeutic targets for colorectal carcinomas.

## 1. Introduction

FAM83H is primarily known for its importance in amelogenesis imperfecta and is involved in enamel formation during teeth development [1,2]. However, the role of FAM83H is not restricted to tooth development. It regulates cell biology in various human cells [3,4,5,6,7]. The expression of FAM83H was elevated in cancer tissue over its nonneoplastic counterpart tissue [4,5,8]. In addition, a higher expression of FAM83H is associated with a poor prognosis in human cancer patients [3,4,7,9]. In cancer of the liver [3], bone [6], kidney [9], and stomach [7], higher expressions of FAM83H predicted shorter survival. FAM83H is involved in the progression of human cancers in combination with various oncogenic signaling molecules. In conjunction with MYC and the Wnt/β-catenin pathway, FAM83H regulates cellular proliferation and invasiveness through the epithelial-to-mesenchymal transition (EMT) [3,6,10]. The role of FAM83H in EMT is suggested to be related to its localization in the cytoplasm and nuclei [7,11]. Deregulated localization of FAM83H in cancer cells has been reported in colorectal carcinoma (CRC) [11], hepatocellular carcinoma [3], gastric carcinomas [7], and gallbladder carcinoma [12].

SCRIB is a polarity protein essential in maintaining the epithelial junction and is deregulated in various pathologic conditions [13,14,15]. The abnormal expression pattern of SCRIB results in the alteration of cellular polarity [13,16]. Because of the deregulation of SCRIB in human cancers, SCRIB has been suggested to be a tumor suppressor [17]. A decreased expression of SCRIB in cancer-associated fibroblasts was associated with shorter survival of lung cancer patients [18]. However, an increasing number of reports have presented SCRIB as an important therapeutic target for cancers. In epithelial cells, the mislocalization of SCRIB resulted in the loss of the E-cadherin-mediated induction of EMT [16,19] and the activation of YAP1 signaling [20,21]. In the liver [22] and breast [23], the deregulation of SCRIB promoted tumor formation. In addition, a higher expression of SCRIB was associated with the shorter survival of ovarian and gastric cancer patients [7,24].

CRC is the third most common cancer, with the second-highest death rate [25]. Globally, 2.2 million new patients and 1.1 million deaths are predicted by 2030 [25]. The etiology and pathogenesis of CRC indicate that CRC is caused by multiple factors, such as lifestyle, genetic disorders, and the immune system [26]. Lifestyle changes are especially continuing to increase CRC incidence worldwide [26]. In addition, despite continuous improvement in the survival of CRC patients, the outcomes of CRCs vary widely according to cancer-specific molecular features and patient characteristics [27]. Therefore, it is important to find adequate biomarkers useful in treating CRCs. Recently, FAM83H and SCRIB have been presented as important regulators of human cancers [3,4,7,15,20], and FAM83H was closely associated with SCRIB in the progression of gastric carcinomas [7]. Furthermore, a search of the public database (GEPIA, http://gepia.cancer-pku.cn, accessed on 25 May 2022) indicated a significant association between SCRIB and FAM83H (Pearson’s correlation *r* = 0.74, *p* < 0.001) in CRC cancer [28]. However, there is no well-established study showing the relationship between FAM83H and SCRIB in CRCs. Therefore, this study aimed to investigate the expression and clinical significance of FAM83H and SCRIB in CRC patients.

## 2. Materials and Methods

### 2.1. Colorectal Carcinoma Patients

This study was approved by the institutional review board of Jeonbuk National University Hospital (IRB number, CUH 2021-10-034) and was performed in compliance with the Declaration of Helsinki. The paraffin-embedded tissue blocks, microscopic slides, and clinicopathologic information for this study were provided by the Biobank of Jeonbuk National University Hospital. Written informed consent was obtained by the Biobank of Jeonbuk National University Hospital. The patients included in this study were CRC patients who had received an operation between January 2018 and May 2019 at Jeonbuk National University Hospital, and their biospecimen were archived in the Biobank of Jeonbuk National University Hospital. Two hundred and twenty-two cases of CRCs were included in this study, and all cases were evaluated according to the latest WHO classification [26] and the American Joint Committee Cancer Staging System [29]. The patients who received neoadjuvant chemotherapy were not included in this study. Eighty-four patients received adjuvant chemotherapy principally based on the FOLFOX regimen (5-fluorouracil, leucovorin, and oxaliplatin) for twelve cycles over six months. The clinicopathological factors included in this study were the age of the patients, sex, site of the tumor, histologic grade, preoperative serum level of CEA (reference value; ~5.0 ng/mL), and CA19-9 (reference value; ~37 kU/L), TNM stage, T category of the stage, lymph node metastasis, and distant metastasis.

### 2.2. Immunohistochemical Staining and Scoring

The expressions of FAM83H and SCRIB in CRC tissue samples were evaluated with immunohistochemical staining. Immunohistochemical staining was performed on tissue microarray sections with 4 μm thickness. The tissue microarrays contained two 3.0 mm cores per case. Antigen retrieval of tissue sections was performed by boiling in pH 6.0 antigen retrieval solution (DAKO, Glostrup, Denmark) for 20 min. The tissue sections were incubated with primary antibodies for FAM83H (dilution: 1:100, Bethyl Laboratories, Montgomery, TX, USA) and SCRIB (dilution: 1:50, Santa Cruz Biotechnology, Santa Cruz, CA, USA). Then, the appropriate secondary antibodies were applied and samples counterstained with hematoxylin. The immunostained slides were evaluated by all authors by consensus under multi-viewing microscopy. The staining slides were scored according to staining intensity (0; no, 1; weak, 2; intermediate, 3; strong) and staining area scores (0; 0%, 1; 0~1%, 2; 2~10%, 3; 11~33%, 4; 34~66%, 5; 67~100%) [6,12,30]. The score of each tissue microarray core was determined by the sum of the intensity score and staining area score. Therefore, the scores ranged from zero to eight. After that, we used the sum of immunohistochemical staining scores from each core. The final immunohistochemical staining score ranged from zero to sixteen [3,9,24,31]

### 2.3. Statistical Analysis

The positivity for the immunohistochemical expression of FAM83H and SCRIB was determined by using a receiver operating characteristic (ROC) curve analysis [32]. The cut-off point for the immunohistochemical staining score was the point with the highest area under the curve (AUC) to predict the cancer-related deaths of patients [6,24,32]. The prognosis of CRC patients was evaluated for cancer-specific survival (CSS) and relapse-free survival (RFS) through September 2021. An event in CSS analysis was the death of a patient from CRC. An event in the RFS analysis was a relapse of CRC or the death of a patient from CRC. Statistical analysis was performed with the chi-square test, Kaplan–Meier survival analysis, and univariate and multivariate Cox proportional hazards regression analysis using SPSS software (IBM, version 25.0, Armonk, NY, USA). Statistical significance was determined by a *p*-value less than 0.05.

## 3. Results

### 3.1. Clinicopathologic Significance of Immunohistochemical Expressions of FAM83H and SCRIB in CRCs

Immunohistochemically, representative examples of FAM83H and SCRIB expression in normal colonic and CRC tissue are presented in Figure 1A. The expression of the mRNA of FAM83H and SCRIB was higher in CRC tissue compared to normal conic tissue in the GEPIA2 database (Figure 1B) [28]. In our human CRC tissue, the expressions of FAM83H and SCRIB were primarily seen in the cytoplasm and nuclei of the cancer cells (Figure 1A). Therefore, we separately evaluated their nuclear and cytoplasmic expressions: nuclear expression of FAM83H (n-FAM83H), cytoplasmic expression of FAM83H (c-FAM83H), nuclear expression of SCRIB (n-SCRIB), and cytoplasmic expression of SCRIB (c-SCRIB). The immunohistochemical expressions of n-FAM83H, c-FAM83H, n-SCRIB, and c-SCRIB were classified into negative or positive groups according to their expression intensity and area. The cut-off points were determined as the points with the highest area under the curve in ROC curve analysis to predict the deaths of the patients (Figure 1C). The cut-off points for the expressions of n-FAM83H, c-FAM83H, n-SCRIB, and c-SCRIB were 5, 14, 8, and 14, respectively (Figure 1C). The cases with immunohistochemical staining scores equal to or greater than 5 for n-FAM83H, 14 for c-FAM83H, 8 for n-SCRIB, and 14 for c-SCRIB expression were considered positive (Figure 1B). With these cut-off values, n-FAM83H positivity was significantly associated with the histologic grade (*p* = 0.042), serum level of CA19-9 (*p* < 0.001), T category of stage (*p* = 0.033), distant metastasis (*p* = 0.003), and the expressions of c-SCRIB (*p* = 0.018) and n-SCRIB (*p* = 0.011) (Table 1). The expression of c-FAM83H was significantly associated with sex (*p* = 0.020) and c-SCRIB expression (*p* < 0.001) (Table 1). The positivity of n-SCRIB was significantly associated with the serum levels of CEA (*p* = 0.011) and CA19-9 (*p* = 0.006), stage (*p* = 0.003), lymph node metastasis (*p* = 0.005), distant metastasis (*p* < 0.001), and the expression of c-SCRIB (*p* < 0.001) (Table 1). The expression of c-SCRIB was significantly associated with distant metastasis (*p* = 0.013) (Table 1).

### 3.2. Individual Expressions of FAM83H and SCRIB Were Associated with the Survival of CRC Patients

The prognoses of CRC patients were evaluated for CSS and RFS. In univariate analysis, the ages of the patients (CSS; *p* = 0.007, RFS; *p* = 0.062), elevated serum levels of CEA (CSS; *p* < 0.001, RFS; *p* < 0.001) and CA19-9 (CSS; *p* < 0.001, RFS; *p* < 0.001), cancer stage (CSS; *p* < 0.001, RFS; *p* < 0.001), T categories of stage (CSS; *p* < 0.001, RFS; *p* < 0.001), lymph node metastasis (CSS; *p* < 0.001, RFS; *p* < 0.001), distant metastasis at diagnosis (CSS; *p* < 0.001, RFS; *p* < 0.001), and the expressions of n-SCRIB (CSS; *p* < 0.001, RFS; *p* = 0.001), c-SCRIB (CSS; *p* < 0.001, RFS; *p* < 0.001), c-FAM83H (CSS; *p* = 0.006, RFS; *p* = 0.307), and n-FAM83H (CSS; *p* < 0.001, RFS; *p* < 0.001) were significantly associated with CSS or RFS (Table 2). The patients positive for n-FAM83H expression had a 6.793-fold (95% confidence interval (95% CI); 2.686–17.179) greater risk of death and a 2.356-fold (95% CI; 11.379–4.026) greater risk of relapse of tumor or death compared to the patients with an n-FAM83H negative tumor (Table 2). The positivity of c-FAM83H showed a 2.291-fold (95% CI; 1.272–4.126) greater risk in CSS (Table 2). The n-SCRIB positivity showed a 3.940-fold (95% CI; 2.213–7.014) greater risk in CSS and a 2.731-fold (95% CI; 1.675–4.453) greater risk in RFS (Table 2). The positivity of c-SCRIB showed a 3.597-fold (95% CI; 1.897–6.819) greater risk in CSS and a 2.175-fold (95% CI; 1.347–3.514) greater risk in RFS (Table 2). The Kaplan–Meier survival curves for CSS and RFS according to the expressions of n-FAM83H, c-FAM83H, n-SCRIB, and c-SCRIB are presented in Figure 2.

Multivariate analysis was performed with the factors significantly associated with CSS and RFS: preoperative serum levels of CEA and CA19-9, tumor stage, T category of tumor stage, lymph node metastasis, distant metastasis, and the expressions of n-FAM83H, n-SCRIB, and c-SCRIB. The factors significantly associated with CSS in multivariate analysis were CA19-9 (*p* = 0.004), T category of stage (*p* = 0.041), distant metastasis (*p* < 0.001), and the expressions of n-SCRIB (*p* = 0.042), c-SCRIB (*p* = 0.033), and n-FAM83H (*p* = 0.020) (Table 3). Distant metastasis was the factor significantly associated with RFS in multivariate analysis (*p* < 0.001) (Table 3). n-FAM83H positivity predicted a 3.170-fold (95% CI; 1.197–8.397) greater death risk than FAM83H-negative tumors (Table 3). c-SCRIB positivity showed a 2.087-fold (95% CI; 1.063–4.099) greater risk in CSS (Table 3). n-SCRIB-positive cancer patients had a 1.884-fold (95% CI; 1.022–3.474) greater risk of CSS (Table 3).

### 3.3. The Co-Expression Pattern of FAM83H and SCRIB Predicted the Survival of CRC Patients

In this study, there were significant associations among the expressions of n-FAM83H, c-FAM83H, n-SCRIB, and c-SCRIB (Table 1). The relationship between FAM83H and SCRIB was also reported in gastric cancers [7]. Based on this relationship, we evaluated the clinical significance of the combined expression patterns of n-FAM83H, c-FAM83H, n-SCRIB, and c-SCRIB in CRCs: n-FAM83H/c-SCRIB, n-FAM83H/n-SCRIB, c-FAM83H/c-SCRIB, and c-FAM83H/n-SCRIB. Among the four types of combined expression patterns, the co-expression pattern of n-FAM83H/c-SCRIB had the largest AUC (AUC = 0.775, *p* < 0.001) (Figure 3A). Therefore, we first evaluated the prognostic significance of the co-expression pattern of n-FAM83H/c-SCRIB in CRC patients. Representative images of the co-expression pattern of n-FAM83H and c-SCRIB in the same sample are presented in Figure 3B. Based on n-FAM83H/c-SCRIB co-expression patterns, the CRCs were grouped into four subgroups: n-FAM83H^−^/c-SCRIB^−^, n = 59; n-FAM83H^−^/c-SCRIB^+^, n = 32; n-FAM83H^+^/c-SCRIB^−^, n = 64; n-FAM83H^+^/c-SCRIB^+^, n = 67. Of these four subgroups, the n-FAM83H^−^/c-SCRIB^−^ subgroup had the longest survival and the n-FAM83H^+^/c-SCRIB^+^ subgroup had the shortest survival (CSS; Log-rank overall *p* < 0.001, RFS; Log-rank overall *p* < 0.001) (Figure 3C). The n-FAM83H^−^/c-SCRIB^−^ subgroup had a 96% three-year CSS rate and an 81% three-year RFS rate (Table 4). In contrast, the n-FAM83H^+^/c-SCRIB^+^ subgroup had only a 53% three-year CSS rate and a 45% three-year RFS rate (Table 4). However, there were no significant prognostic differences among the n-FAM83H^−^/c-SCRIB^−^, n-FAM83H^−^/c-SCRIB^+^, and n-FAM83H^+^/c-SCRIB^−^ subgroups (Figure 3C). Therefore, we further classified the CRC patients into two prognostic subgroups: (n-FAM83H^−^/c-SCRIB^−^, n-FAM83H^−^/c-SCRIB^+^, or n-FAM83H^+^/c-SCRIB^−^) and (n-FAM83H^+^/c-SCRIB^+^) subgroups (Figure 3D). This subgrouping into two prognostic subgroups was significantly associated with CSS and RFS in univariate (CSS; *p* < 0.001, RFS; *p* < 0.001) and multivariate analysis (CSS; *p* < 0.001, RFS; *p* = 0.032) (Table 5). In multivariate analysis, the n-FAM83H^+^/c-SCRIB^+^ subgroup had a 3.210-fold (95% CI; 1.683–6.122) greater risk of shorter CSS and a 1.710-fold (95% CI; 1.047–2.793) greater risk of shorter RFS (Table 5). In addition, when we evaluated the prognostic significance of the co-expression pattern of n-FAM83H/n-SCRIB, subgrouping into the two prognostic subgroups was significantly associated with CSS (*p* < 0.001) and RFS (*p* < 0.001) in univariate analysis (Table 5). In multivariate analysis, the n-FAM83H^+^/n-SCRIB^+^ subgroup had a 2.847-fold (95% CI; 1.537–5.272, *p* < 0.001) greater risk of shorter CSS (Table 5).

### 3.4. Individual and Co-Expression Patterns of FAM83H and SCRIB Were Associated with Survival in the Subpopulation of CRCs According to Therapeutic Treatment

Based on the prognostic significance of FAM83H and SCRIB expressions in CRC, we further evaluated their prognostic significance in the subpopulation of CRCs according to adjuvant chemotherapy. In the 138 patients who did not receive adjuvant chemotherapy, the expressions of n-FAM83H (CSS; *p* = 0.002, RFS; *p* = 0.011), c-FAM83H (CSS; *p* = 0.021, RFS; *p* = 0.037), n-SCRIB (CSS; *p* < 0.001, RFS; *p* < 0.001), and c-SCRIB (CSS; *p* = 0.003, RFS; *p* = 0.007) and the co-expression pattern of n-FAM83H/c-SCRIB (CSS; *p* < 0.001, RFS; *p* < 0.001) were significantly associated with CSS and RFS (Figure 4A). In the 84 patients who received adjuvant chemotherapy, the expressions of n-FAM83H (*p* < 0.001), n-SCRIB (*p* < 0.001), and c-SCRIB (*p* = 0.006) and the co-expression pattern of n-FAM83H/c-SCRIB (*p* < 0.001) were significantly associated with CSS (Figure 4B). However, only the co-expression pattern of n-FAM83H/c-SCRIB (*p* = 0.009) was significantly associated with RFS (Figure 4B).

## 4. Discussion

In this study, we have shown that the expressions of FAM83H and SCRIB were associated with advanced clinicopathologic characteristics of CRC patients. The expression of n-FAM83H was associated with a higher histologic grade, increased serum levels of CA19-9, a higher T category, and distant metastasis. In addition, higher expressions of n-FAM83H and c-FAM83H were associated with a shorter CSS of CRC patients in univariate analysis. In multivariate analysis, the expression of n-FAM83H predicted shorter CSS in CRC patients. Consistently, a higher expression of FAM83H in cancer tissue was reported in cancers of the breast, colon, liver, lung, ovary, pancreas, stomach, and uterus [4,5,8]. FAM83H positivity was associated with a higher uterine cancer stage [8] and pancreatic ductal carcinoma [33]. In line with our results, a higher expression of FAM83H was associated with the poor prognosis of cancer patients with tumors of the bladder [31], bone [6], gallbladder [12], kidney [9], liver [3], lung [34], stomach [7], and uterus [4]. However, in contrast to our results, there have been controversial reports on the significance of the expression of FAM83H in human cancers. The expression of FAM83H was decreased in cutaneous squamous cell carcinomas compared to normal tissue [35], and a higher expression of FAM83H was associated with a favorable prognosis in gastric adenocarcinomas [36]. Therefore, despite the prognostic significance of FAM83H expression in CRC patients, further study is needed to clarify the clinical relevance of FAM893H expression in CRC.

In addition to the prognostic significance of n-FAM83H expression, the expression of SCRIB was also significantly associated with shorter survival of CRC patients. The expression of n-SCRIB was significantly associated with higher serum levels of CEA and CA19-9, a higher tumor stage, lymph node metastasis, and distant metastasis. In addition, higher expressions of n-SCRIB and c-SCRIB were associated with a shorter CSS of CRC patients in univariate and multivariate analysis. In line with our results, a higher expression of SCRIB in human cancers has been reported in CRC [20,37], hepatocellular carcinoma [22], breast cancer [15], and prostate cancer [15]. Furthermore, a higher expression of SCRIB also predicted shorter survival of ovarian carcinoma [24] and gastric carcinoma [7]. However, controversially, the expression of SCRIB was decreased compared to normal tissue in oropharyngeal squamous cell carcinoma, breast cancer, uterine cervical cancer, lung cancer, and lymphoma [15,38]. Therefore, although our results suggest that the expression of SCRIB might be used as a prognostic indicator for CRC patients, further study is needed to determine the clinical significance of SCRIB expression.

In this study, there was a significant association between the expressions of FAM83H and SCRIB, as well as their co-expression patterns, which were independent prognostic indicators of CSS and RFS in CRC patients. The molecular relationship between FAM83H and SCRIB is evident in data from the public database [28] and is further supported by a study on the relationship between FAM83H and SCRIB in gastric carcinomas [7]. In gastric carcinomas, FAM83H is involved in the regulation of the expression of SCRIB and forms the FAM83H-SCRIB complex, and FAM83H-SCRIB stabilizes β-catenin to induce EMT and the proliferation of cancer cells [7]. FAM83H overexpression especially activated the tumor growth and pulmonary metastasis of gastric cancer cells, which was attenuated with the knock-down of SCRIB in vivo [7]. These results suggest that the FAM83H-mediated activation of SCRIB might be an important mechanism in FAM83H-mediated oncogenesis. This relationship between FAM83H and SCRIB might be explained by the shorter survival of CRC patients with an n-FAM83H^+^/c-SCRIB^+^ phenotype. In gastric carcinomas, patients with positive expressions of both n-FAM83H and n-SCRIB showed the shortest overall survival and RFS [7]. In addition to the molecular association of FAM83H and SCRIB in human cancers, FAM83H forms a more extensive molecular network in the progression of human cancers. In hepatocellular carcinomas, oncogene *MYC* induces FAM83H transcription, and the increased expression of FAM83H stimulates the growth and invasiveness of cells [3]. In osteosarcoma cells, FAM83H activated the canonical Wnt/β-catenin pathway by stabilizing β-catenin from proteasome-mediated ubiquitination [6]. A higher expression of FAM83H in KHOS/NP osteosarcoma cells increased in vivo tumor growth and pulmonary metastasis [6]. In clear cell renal cell carcinomas, the co-expression of FAM83H and pannexin-2 predicted the shortest overall survival and RFS in multivariate analysis [9]. In addition, FAM83H is involved in the proliferation of cancer cells by activating PI3K/AKT pathway in pancreatic cancer [10] and uterine cervical cancer [5]. Therefore, it is suggested that FAM83H is involved in the progression of human cancers by forming a complex network with the molecules associated with cancer progression and SCRIB in CRCs.

With regard to the prognostic significance of FAM83H and SCRIB expression in CRC patients, an interesting finding in this study is that the co-expression pattern of n-FAM83H and c-SCRIB was associated with the CSS and RFS in the patients who received adjuvant chemotherapy. This result suggests that higher expressions of FAM83H and/or SCRIB would be associated with resistance to chemotherapy. In addition, the positivity of n-FAM83H or n-SCRIB was associated with a higher tumor stage, lymph node metastasis, or distant metastasis at diagnosis. Invasive and metastatic potential is an important clinical feature of EMT, and EMT has been presented as an important phenotype of resistance to conventional anti-cancer therapies [39,40]. These results suggest the possibility that FAM83H/SCRIB might be involved in resistance to anti-cancer chemotherapy through the induction of EMT. Similarly, a higher expression of n-FAM83H and n-SCRIB was associated with higher tumor stage and lymph node metastasis of gastric carcinomas [7]. EMT phenotype, loss of E-cadherin with the induction of N-cadherin and Snail has been induced with the overexpression of FAM83H and SCRIB in gastric cancer cells and ovarian cancer cells [7,24]. A higher expression of FAM83H was associated with the induction of EMT by stimulating the PI3K pathway in pancreatic cancer cells [10]. In ovarian carcinomas, the positivity of n-SCRIB was significantly associated with platinum resistance and has been suggested as an independent indicator of poor prognosis for ovarian carcinoma patients who received adjuvant chemotherapy [24]. In in vivo experiments, the overexpression of FAM83H increased the pulmonary metastasis of NCI-N87 gastric cancer cells [7] and KHOS/NP osteosarcoma cells [6]. Regarding therapeutic application targeting the FAM83H-SCRIB pathway, the knock-down of FAM83H and/or SCRIB inhibited the growth of various human cancer cells, such as hepatocellular carcinoma [3], gastric carcinoma [7], ovarian carcinoma [24], and osteosarcoma [6]. Therefore, the FAM83H-SCRIB pathway might be a therapeutic target for human cancers, especially for the poorly prognostic subgroup highly expressing FAM83H and SCRIB. However, further study, especially on the precise mechanism regarding the role of FAM83H-SCRIB in conventional anti-cancer therapies, is needed to select a population of patients who could potentially benefit from anti-FAM83H/SCRIB-targeted therapy.

Our results suggest that the expressions of FAM83H and SCRIB might be used as prognostic markers for CRC patients. With regard to the evaluation of immunohistochemically stained slides, scoring according to the subcellular localization of selected markers is an important point. In evaluating the subcellular localization of FAM83H and SCRIB, FAM83H expression is expected in the cytoplasmic membrane and the cytosol [11,41,42], and SCRIB expression is expected in the plasma membrane [13,14]. However, as shown in Figure 1A, the expressions of FAM83H and SCRIB are primarily seen in the cytoplasm and nuclei. Furthermore, the nuclear expressions of FAM83H and SCRIB were independent indicators of shorter CSS in CRC patients. Consistently, n-FAM83H expression was an indicator of the poor prognosis of hepatocellular carcinoma [3], gastric carcinoma [7], clear cell renal cell carcinoma [9], and osteosarcoma [6]. In gastric carcinoma and ovarian carcinomas, positivity for n-SCRIB was also significantly associated with the shorter survival of patients [7,24]. The prognostic significance of the nuclear expressions of FAM83H and SCRIB might be related to their roles in cancer progression in conjunction with nuclear proteins important in cancer progression, such as MYC and β-catenin [3,7,24]. In CRC, the nuclear localization of FAM83H was present in a minor population of cancers, but its localization to nuclei has been suggested to be involved in cancer progression with the interaction of SON and casein kinase 1α [11,42]. Therefore, our results suggest that careful evaluation of the subcellular localization of FAM83H and SCRIB is important for the FAM83H and SCRIB immunostaining of cancer tissue. However, further study is needed to clarify the significance of the expressions of FAM83H and SCRIB in predicting the prognosis of human cancers.

## 5. Conclusions

In conclusion, the results show that individual and combined expression patterns of FAM83H and SCRIB are significantly associated with shorter CSS and RFS in CRC patients. The combined expression pattern of n-FAM83H and c-SCRIB was especially significantly associated with the shorter survival of CRC patients who received adjuvant chemotherapy. Therefore, FAM83H and SCRIB might be used as prognostic markers for CRC patients and as potential therapeutic targets for the poor prognostic population of CRC patients who have tumors that highly express FAM83H and SCRIB.

## Figures and Tables

**Figure 1 diagnostics-12-01579-f001:**
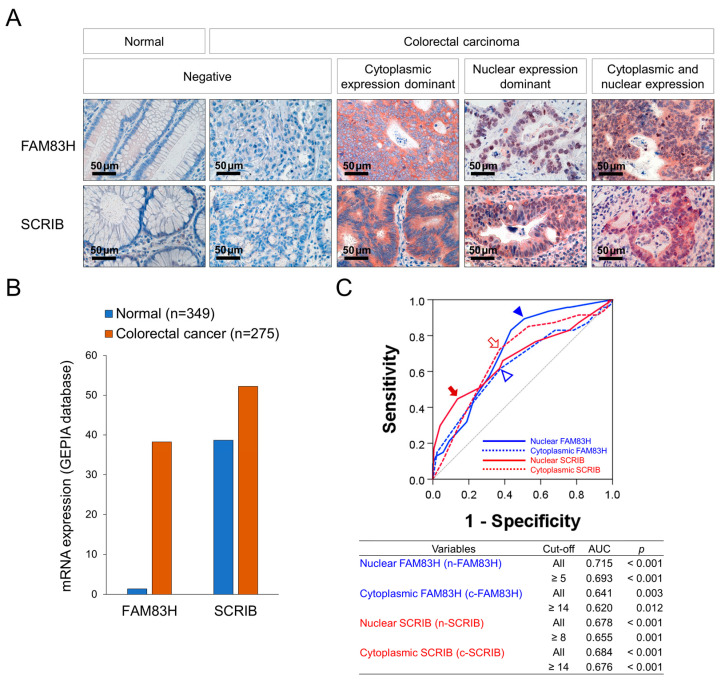
Immunohistochemical staining for FAM83H and SCRIB and statistical analysis in CRCs. (**A**) Immunohistochemical expressions of FAM83H and SCRIB in normal and CRC tissue. FAM83H and SCRIB were expressed in the cytoplasm and nuclei of the CRC cells. (**B**) The expression of the mRNA of FAM83H and SCRIB in normal colonic tissue and colorectal carcinoma tissue from the GEPIA2 database (http://gepia2.cancer-pku.cn, accessed on 23 June 2022) (**C**) Receiver operating characteristic curve analysis was performed to determine the positivity of the nuclear expression of FAM83H (n-FAM83H), cytoplasmic expression of FAM83H (c-FAM83H), nuclear expression of SCRIB (n-SCRIB), and cytoplasmic expression of SCRIB (c-SCRIB). The blue arrowhead indicates the cut-off point for n-FAM83H expression, and the empty blue arrowhead indicates the cut-off point for c-SCRIB expression. The red arrow indicates the cut-off point for n-SCRIB expression, and the open red arrow indicates the cut-off point for c-SCRIB expression. The cut-off points were determined to predict the deaths of CRC patients, and the points have the highest area under the curve (AUC).

**Figure 2 diagnostics-12-01579-f002:**
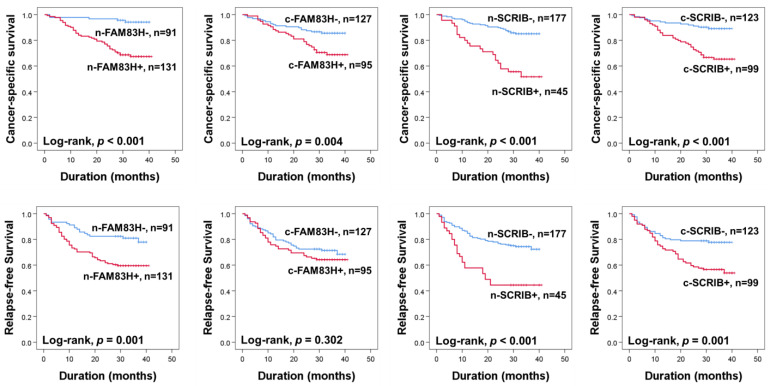
Kaplan–Meier survival analysis for cancer-specific survival and relapse-free survival according to nuclear and cytoplasmic expressions of FAM83H and SCRIB in CRC patients. n-FAM83H, nuclear expression of FAM83H; c-FAM83H, cytoplasmic expression of FAM83H; n-SCRIB, nuclear expression of SCRIB; c-SCRIB, cytoplasmic expression of SCRIB.

**Figure 3 diagnostics-12-01579-f003:**
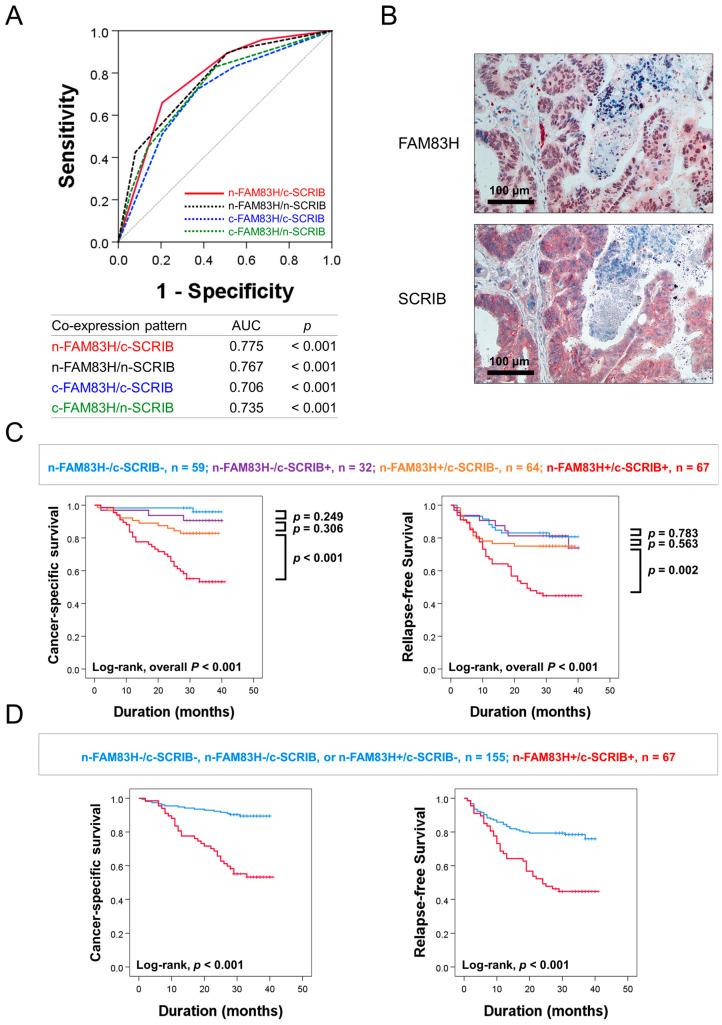
Statistical analysis and Kaplan–Meier survival analysis according to the combined expression pattern of nuclear and cytoplasmic expressions of FAM83H and SCRIB. (**A**) Receiver operating characteristic curve analysis to predict the death of CRC patients according to combined expression patterns of nuclear and cytoplasmic expressions of FAM83H and SCRIB: n-FAM83H/c-SCRIB, n-FAM83H/n-SCRIB, c-FAM83H/c-SCRIB, and c-FAM83H/n-SCRIB. (**B**) Representative images of the co-expression patterns of n-FAM83H and c-SCRIB in the same CRC sample. (**C**) Kaplan–Meier survival analysis in four prognostic subgroups of CRCs according to the combined expression pattern of n-FAM83H and c-SCRIB: (n-FAM83H^−^/c-SCRIB^−^), (n-FAM83H^−^/c-SCRIB^+^), (n-FAM83H^+^/c-SCRIB^−^), and (n-FAM83H^+^/c-SCRIB^+^) subgroups. (**D**) Survival analysis in two prognostic subgroups of CRCs: (n-FAM83H^−^/c-SCRIB^−^, n-FAM83H^−^/c-SCRIB^+^, or n-FAM83H^+^/c-SCRIB^−^) and (n-FAM83H^+^/c-SCRIB^+^) subgroups.

**Figure 4 diagnostics-12-01579-f004:**
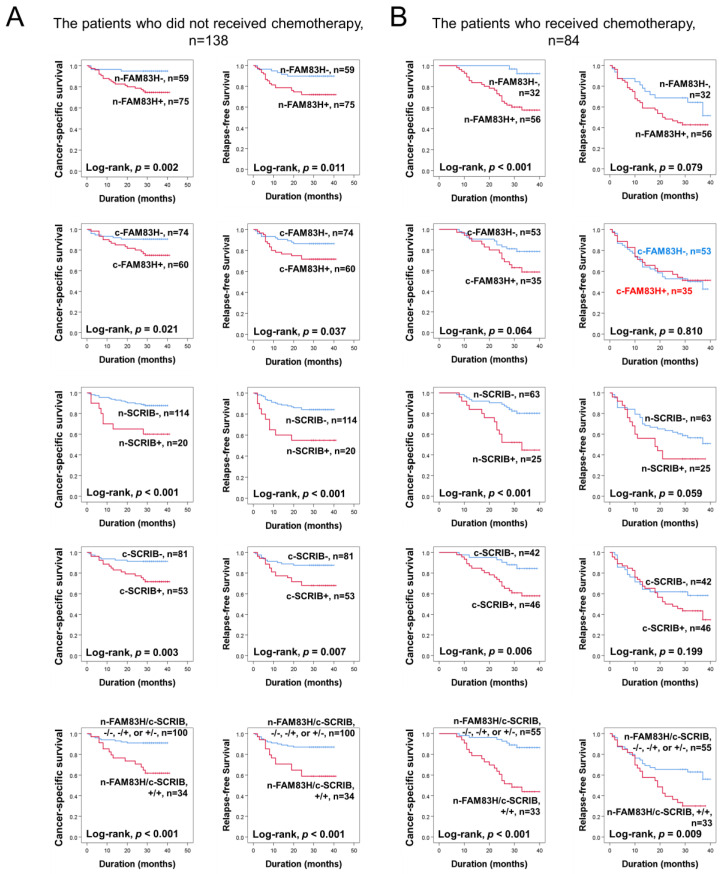
Survival analysis according to individual expressions of nuclear FAM83H (n-FAM83H), cytoplasmic FAM83H (c-FAM83H), nuclear SCRIB (n-SCRIB), cytoplasmic FAM83H (c-SCRIB), and co-expression patterns of nuclear FAM83H and cytoplasmic SCRIB (n-FAM83H/c-SCRIB) in the subgroups of CRC patients who received or did not receive adjuvant chemotherapy. (**A**) Kaplan–Meier survival curves in the 138 patients who did not receive chemotherapy. (**B**) Kaplan–Meier survival curves in the 84 patients who received adjuvant chemotherapy.

**Table 1 diagnostics-12-01579-t001:** Association between the expressions of FAM83H and SCRIB with clinicopathological characteristics of 122 CRCs.

Characteristics		No.	n-FAM83H		c-FAM83H		n-SCRIB		c-SCRIB	
			Positive	*p*	Positive	*p*	Positive	*p*	Positive	*p*
Age	<70	101	60 (59%)	0.913	42 (42%)	0.739	22 (22%)	0.609	47 (47%)	0.595
	≥70	121	71 (59%)		53 (44%)		23 (19%)		52 (43%)	
Sex	Female	97	58 (60%)	0.834	50 (52%)	**0.020**	2 (2%)	0.909	45 (46%)	0.635
	Male	125	73 (58%)		45 (36%)		25 (20%)		54 (43%)	
Tumor site	Left *	150	82 (55%)	0.058	61 (41%)	0.355	29 919%)	0.616	67 (45%)	0.975
	Right **	72	49 (68%)		34 (47%)		16 (22%)		32 (44%)	
Histologic grade	WD, MD	205	117 (57%)	**0.042**	91 (44%)	0.095	41 (20%)	0.728	93 (45%)	0.422
	PD	17	14 (82%)		4 (24%)		4 (24%)		6 (35%)	
CEA	≤5.0 ng/mL	166	94 (57%)	0.214	77 (46%)	0.062	27 (16%)	**0.011**	73 (44%)	0.750
	>5.0 ng/mL	56	37 (66%)		18 (32%)		18 (32%)		26 (46%)	
CA19-9	≤37 kU/L	191	103 (54%)	**<0.001**	81 (42%)	0.774	33 (17%)	**0.006**	82 (43%)	0.216
	>37 kU/L	31	28 (90%)		14 (45%)		12 (39%)		17 (55%)	
Stage	I and II	118	64 (54%)	0.124	47 (40%)	0.342	15 (13%)	**0.003**	47 (40%)	0.128
	III and IV	104	67 (64%)		48 (46%)		30 (29%)		52 (50%)	
T category	T1, T2, T3	186	104 (56%)	**0.033**	81 (44%)	0.605	34 (18%)	0.094	79 (42%)	0.148
	T4	36	27 (75%)		14 (39%)		11 (31%)		20 (56%)	
LN metastasis	Absence	125	69 (55%)	0.190	49 (39%)	0.219	17 (14%)	**0.005**	50 (40%)	0.118
	Presence	97	62 (64%)		46 (47%)		28 (29%)		49 (51%)	
Distant metastasis	Absence	170	91 (54%)	**0.003**	73 (43%)	0.936	24 (14%)	**<0.001**	68 (40%)	**0.013**
	Presence	52	40 (77%)		22 (42%)		21 (40%)		31 (60%)	
c-SCRIB	Negative	123	64 (52%)	**0.018**	35 (28%)	**<0.001**	15 (12%)	**<0.001**		
	Positive	99	67 (68%)		60 (61%)		30 (30%)			
n-SCRIB	Negative	177	97 (55%)	**0.011**	74 (42%)	0.556				
	Positive	45	34 (76%)		21 (47%)					
c-FAM83H	Negative	127	73 (57%)	0.592						
	Positive	95	58 (61%)							

Abbreviations: WD, well differentiated; MD, moderately differentiated; PD, poorly differentiated; CEA, carcinoembryonic antigen; CA19-9, carbohydrate antigen 19-9; LN, lymph node; n-FAM83H, nuclear expression of FAM83H; c-FAM83H, cytoplasmic expression of FAM83H; n-SCRIB, nuclear expression of SCRIB; c-SCRIB, cytoplasmic expression of SCRIB. * Left colon is defined as the segment from the distal one-third of the transverse colon to the rectum. ** Right colon is defined as the segment from the cecum to the proximal two-thirds of the transverse colon.

**Table 2 diagnostics-12-01579-t002:** Univariate Cox proportional hazards regression analysis for cancer-specific survival and relapse-free survival in 122 CRCs.

Characteristics	No.	CSS	*p*	RFS	*p*
		HR (95% CI)		HR (95% CI)	
Age, years, ≥70 (vs. <70)	121	2.408 (1.270–4.563)	**0.007**	1.583 (0.977–2.566)	0.062
Sex, male (vs. female)	125	0.736 (0.415–1.034)	0.293	1.030 (0.643–1.651)	0.901
Tumor site, right (vs. left)	72	1.614 (0.905–2.878)	0.105	1.118 (0.684–1.829)	0.656
Histologic grade, PD (vs. WD and MD)	17	2.159 (0.916–5.088)	0.078	1.536 (0.703–3.354)	0.281
CEA, >5.0 ng/mL (vs. ≤5.0 ng/mL)	56	2.780 (1.563–4.944)	**<0.001**	2.537 (1.577–4.080)	**<0.001**
CA19-9, >37 kU/L (vs. ≤37 kU/L)	31	6.895 (3.839–12.385)	**<0.001**	4.606 (2.784–7.622)	**<0.001**
Stage, III and IV (vs. I and II)	104	4.239 (2.157–8.332)	**<0.001**	6.974 (3.813–12.758)	**<0.001**
T category, T4 (vs. T1–T3)	36	4.187 (2.320–7.556)	**<0.001**	3.873 (2.359–6.358)	**<0.001**
LN metastasis, presence (vs. absence)	97	4.349 (2.256–8.383)	**<0.001**	4.260 (2.536–7.159)	**<0.001**
Distant metastasis, presence (vs. absence)	52	9.678 (5.220–17.944)	**<0.001**	17.804 (10.460–30.305)	**<0.001**
c-SCRIB, positive (vs. negative)	99	3.597 (1.897–6.819)	**<0.001**	2.175 (1.347–3.514)	**0.001**
n-SCRIB, positive (vs. negative)	45	3.940 (2.213–7.014)	**<0.001**	2.731 (1.675–4.453)	**<0.001**
c-FAM83H, positive (vs. negative)	95	2.291 (1.272–4.126)	**0.006**	1.275 (0.800–2.031)	0.307
n-FAM83H, positive (vs. negative)	131	6.793 (2.686–17.179)	**<0.001**	2.356 (1.379–4.026)	**0.002**

Abbreviations: CSS, cancer-specific survival; RFS, relapse-free survival; HR, hazard ratio; 95% CI, 95% confidence interval; CEA, carcinoembryonic antigen; CA19-9, carbohydrate antigen 19-9; LN, lymph node; n-FAM83H, nuclear expression of FAM83H; c-FAM83H, cytoplasmic expression of FAM83H; n-SCRIB, nuclear expression of SCRIB; c-SCRIB, cytoplasmic expression of SCRIB.

**Table 3 diagnostics-12-01579-t003:** Multivariate Cox regression analysis for cancer-specific survival and relapse-free survival.

Characteristics	CSS	*p*	RFS	*p*
	HR (95% CI)		HR (95% CI)	
CA19-9, >37 kU/L (vs. ≤37 kU/L)	2.677 (1.377–5.203)	**0.004**		
T category, T4 (vs. T1–T3)	1.947 (1.028–3.686)	**0.041**		
Distant metastasis, presence (vs. absence)	4.110 (2.009–8.411)	**<0.001**	17.804 (10.460–30.305)	**<0.001**
n-SCRIB, positive (vs. negative)	1.884 (1.022–3.474)	**0.042**		
c-SCRIB, positive (vs. negative)	2.087 (1.063–4.099)	**0.033**		
n-FAM83H, positive (vs. negative)	3.170 (1.197–8.397)	**0.020**		

Abbreviations: CSS, cancer-specific survival; RFS, relapse-free survival; HR, hazard ratio; 95% CI, 95% confidence interval; n-FAM83H, nuclear expression of FAM83H; n-SCRIB, nuclear expression of SCRIB; c-SCRIB, cytoplasmic expression of SCRIB. The variables considered in the multivariate analysis were the preoperative serum levels of CEA and CA19-9, tumor stage, T category of tumor stage, lymph node metastasis, distant metastasis, and the expressions of n-FAM83H, n-SCRIB, and c-SCRIB.

**Table 4 diagnostics-12-01579-t004:** One- and three-year cancer-specific survival and relapse-free survival according to co-expression patterns of n-FAM83H and c-SCRIB.

Co-Expression Pattern of n-FAM83H and c-SCRIB	No.	1y-CSS (%)	3y-CSS (%)	1y-RFS (%)	3y-RFS (%)
Co-expression Model 1					
n-FAM83H^−^/c-SCRIB^−^	59	98	96	88	81
n-FAM83H^−^/c-SCRIB^+^	32	97	91	91	81
n-FAM83H^+^/c-SCRIB^−^	64	91	83	78	75
n-FAM83H^+^/c-SCRIB^+^	67	81	53	67	45
Co-expression Model 2					
n-FAM83H^−^/c-SCRIB^−^, n-FAM83H^−^/c-SCRIB^+^, or n-FAM83H^+^/c-SCRIB^−^	155	95	89	85	78
n-FAM83H^+^/c-SCRIB^+^	67	81	53	67	45

Abbreviations: 1y-CSS, cancer-specific survival rate at one year; 3y-CSS, cancer-specific survival rate at three years; 1y-RFS, relapse-free survival rate at one year; 3y-RFS, relapse-free survival rate at three years; n-FAM83H, nuclear expression of FAM83H; c-SCRIB, cytoplasmic expression of SCRIB.

**Table 5 diagnostics-12-01579-t005:** Univariate and multivariate Cox regression analysis for cancer-specific survival and relapse-free survival according to the co-expression patterns of FAM83H and SCRIB in CRCs.

Characteristics		No.	CSS		RFS	
			HR (95% CI)	*p*	HR (95% CI)	*p*
Univariate analysis						
Co-expression pattern of n-FAM83H and c-SCRIB	+/+ (vs. −/−, −/+, +/−)	67/222	5.387 (2.942–9.862)	**<0.001**	2.923 (1.832–4.662)	**<0.001**
Co-expression pattern of n-FAM83H and n-SCRIB	+/+ (vs. −/−, −/+, +/−)	34/222	5.519 (3.083–9.879)	**<0.001**	3.368 (2.028–5.592)	**<0.001**
Multivariate analysis Model 1						
Co-expression pattern of n-FAM83H and c-SCRIB	+/+ (vs. −/−, −/+, +/−)	67/222	3.210 (1.683–6.122)	**<0.001**	1.710 (1.047–2.793)	**0.032**
Multivariate analysis Model 2						
Co-expression pattern of n-FAM83H and n-SCRIB	+/+ (vs. −/−, −/+, +/−)	34/222	2.847 (1.537–5.272)	**<0.001**		

Abbreviations: CSS, cancer-specific survival; RFS, relapse-free survival; HR, hazard ratio; 95% CI, 95% confidence interval; n-FAM83H, nuclear expression of FAM83H; c-SCRIB, cytoplasmic expression of SCRIB. The variables considered in the multivariate analysis Model 1 were the preoperative serum levels of CEA and CA19-9, tumor stage, T category of the tumor stage, lymph node metastasis, distant metastasis, and the co-expression pattern of n-FAM83H and c-SCRIB. The variables considered in the multivariate analysis Model 2 were the preoperative serum levels of CEA and CA19-9, tumor stage, T category of the tumor stage, lymph node metastasis, distant metastasis, and the co-expression pattern of n-FAM83H and n-SCRIB.

## Data Availability

The datasets generated during and/or analyzed during the current study are available from the corresponding author upon reasonable request.

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
