# Peer review of "Individual and Co-Expression Patterns of FAM83H and SCRIB at Diagnosis Are Associated with the Survival of Colorectal Carcinoma Patients"

_diagnostics, 2022, doi:10.3390/diagnostics12071579_

Round 1

Reviewer 1 Report

The authors address a topic of interest and the performed experiments are appropriate.

I suggest to add the thickness of the tissue sections in material and methods and the scale bar in figure 1. 

The authors could bold the signifcant p Values in the tables.

The authors could add in figure 3 IHC images that represent the co-expression pattern of nuclear FAM83H and cytoplasmic SCRIB in the same sample.

Author Response

Response to reviewer 1

We thank the reviewer for the insightful comments.

Comments to Author:

Reviewer #1:

The authors address a topic of interest and the performed experiments are appropriate.

I suggest to add the thickness of the tissue sections in material and methods and the scale bar in figure 1. 

We thank the reviewer for this comment. In response to the reviewer’s comment, we have added the thickness of the tissue sections in material and methods and the scale bar in Figure 1A and Figure 3B. Below are the revised sentence in the Materials and Methods section and revised Figure 1. 

Materials and Methods

Immunohistochemical staining was performed on tissue microarray sections with 4 μm thickness.

Figure 1A

The authors could bold the significant p Values in the tables.

We thank the reviewer for this comment. In response to the reviewer’s comment, we have bolded the significant p Values.

The authors could add in figure 3 IHC images that represent the co-expression pattern of nuclear FAM83H and cytoplasmic SCRIB in the same sample.

We thank the reviewer for this comment. In response to the reviewer’s comment, we have presented images that represent the co-expression pattern of nuclear FAM83H and cytoplasmic SCRIB in the same sample in Figure 3B. Below are the revised Figure 3B. 

Figure 3B

Reviewer 2 Report

Dear Editor in Chief

The attached manuscript entitled "Individual and co-expression patterns of FAM83H and SCRIB at diagnosis are associated with the survival of colorectal carcinoma patients" presents a new finding on the possible use of FAM83H and SCRIB expression as prognostic markers for the survival of CRC.

The manuscript is well written and the experimental approach is very convincing.

Author Response

Response to reviewer 2

We thank the reviewer for the insightful comments.

Comments to Author:

Reviewer #2:

The attached manuscript entitled "Individual and co-expression patterns of FAM83H and SCRIB at diagnosis are associated with the survival of colorectal carcinoma patients" presents a new finding on the possible use of FAM83H and SCRIB expression as prognostic markers for the survival of CRC.

The manuscript is well written and the experimental approach is very convincing.

We very thank the reviewer for this comment.

Reviewer 3 Report

In this manuscript, Jeong et al demonstrated the feasibility of using FAM83H and SCRIB as prognostic biomarkers for colon cancer. They employed immunohistochemistry to determine the protein expression of FAM83H and SCRIB in about 200 colon cancer tissues. They scored the expression of the target proteins in both cytoplasm and nucleus based on the intensity and percentage of tissue with the staining. The methodology and statistical analysis are validated. Before recommending it for publication, the authors should clarify some issues.

1.     The authors mentioned that CRC patients received adjuvant chemotherapy in the method and material section. Would the authors give more detail on it? For example, the drug and treatment duration.

2.     What are FAM83H and SCR1B expression levels in normal colon tissue?

3.     The scale bar in Figure 1A is missing.

4.     In Figure 1B, what does " ALL " mean in the Cut-off column?

5.     In Figures 3B and 3C, how about the performance of n-FAM83H+ and n-SCRIB+?

6.     In Figures 4A and 4B, which of the survival analyses do they perform in the two panels?

7.     In Figure 4, the authors demonstrated that the expression of FAM83H and SCRIB would have a different prognostic effect on the patients with or without chemotherapy. The authors might need to determine if the high expression of FAM83H and/or SCRIB would be associated with the resistance to chemotherapy.

8.     The authors might need to explain in more detail why they chose n-FAM83H and c-SCRIB. Based on Figure 3A, the performance of n-FAM83H / c-SCRIB and n-FAM83H / n-SCRIB should be similar.     

Author Response

Response to reviewer 3

We thank the reviewer for the insightful comments.

Comments to Author:

Reviewer #3: In this manuscript, Jeong et al demonstrated the feasibility of using FAM83H and SCRIB as prognostic biomarkers for colon cancer. They employed immunohistochemistry to determine the protein expression of FAM83H and SCRIB in about 200 colon cancer tissues. They scored the expression of the target proteins in both cytoplasm and nucleus based on the intensity and percentage of tissue with the staining. The methodology and statistical analysis are validated. Before recommending it for publication, the authors should clarify some issues.

We very thank the reviewer for this comment.

  1. The authors mentioned that CRC patients received adjuvant chemotherapy in the method and material sectionWould the authors give more detail on it? For example, the drug and treatment duration.

We thank the reviewer for this comment. In response to the reviewer’s comment, we have described the drug and treatment duration for adjuvant chemotherapy in the Materials and Methods section. Below is the revised sentence in the Materials and Methods section.

Eighty-four patients received adjuvant chemotherapy principally based on the FOLFOX regimen (5-fluorouracil, leucovorin, and oxaliplatin) for twelve cycles over six months.

  1. What are FAM83H and SCR1B expression levels in normal colon tissue?

We thank the reviewer for this comment. In our tissue microarray cores, some cases included non-neoplastic colorectal mucosa, and the staining for FAM83H and SCRIB was negative or weak. Therefore, in response to the reviewer’s comment, we have added examples of immunohistochemical images in normal colorectal tissue in Figure 1A. In addition, we have presented the data on the expression of mRNA of FAM83H and SCRIB in normal colonic tissue and colorectal carcinoma tissue based on the GEPIA2 database in Figure 1B. Below is the revised Figure 1. 

Figure 1

  1. The scale bar in Figure 1A is missing.

We thank the reviewer for this comment. In response to the reviewer’s comment, we have added the scale bar in Figure 1A and Figure 3B.

  1. In Figure 1B, what does " ALL " mean in the Cut-off column?

We thank the reviewer for this comment. ALL in the cut-off column means that the area under the polygon according to the immunohistochemical staining score (the area under the blue curve in below figure in case of n-FAM83H score). Each vertex of a polygon means one immunohistochemical staining score. In the case of the n-FAM83H score, the cut-off point of the immunohistochemical staining score was five, and AUC in this cut-off point means the area under the red-lined curve in the below figure.

  1. In Figures 3B and 3C, how about the performance of n-FAM83H+ and n-SCRIB+?

We thank the reviewer for this comment. As we described in our manuscript, because the co-expression pattern of n-FAM83H/c-SCRIB had the largest AUC (AUC = 0.775, p < 0.001) among the four types of combined expression patterns, we have presented data for the co-expression pattern of n-FAM83H/c-SCRIB. Furthermore, in response to the reviewer’s comment, we have presented the data of the co-expression pattern of n-FAM83H+ and n-SCRIB+ in the results section and Table 5. Below are the revised sentences in the results section and revised Table 5.

Results section

In addition, when we evaluated the prognostic significance of the co-expression pattern of n-FAM83H/n-SCRIB, subgrouping into two prognostic subgroups was significantly associated with CSS (p < 0.001) and RFS (p < 0.001) in univariate analysis (Table 5). In multivariate analysis, the n-FAM83H+/n-SCRIB+ subgroup had a 2.847-fold (95% CI; 1.537-5.272, p < 0.001) greater risk of shorter CSS (Table 5).

Table 5. Univariate and multivariate Cox regression analysis for cancer-specific survival and relapse-free survival according to the co-expression patterns of FAM83H and SCRIB in CRCs.

Characteristics

No.

CSS

RFS

HR (95% CI)

p

HR (95% CI)

p

Univariate analysis

Coexpression pattern of n-FAM83H and c-SCRIB

+/+ (vs. -/-, -/+, +/-)

67/222

5.387 (2.942-9.862)

< 0.001

2.923 (1.832-4.662)

< 0.001

Coexpression pattern of n-FAM83H and n-SCRIB

+/+ (vs. -/-, -/+, +/-)

34/222

5.519 (3.083-9.879)

< 0.001

3.368 (2.028-5.592)

< 0.001

Multivariate analysis Model 1

Coexpression pattern of n-FAM83H and c-SCRIB

+/+ (vs. -/-, -/+, +/-)

67/222

3.210 (1.683-6.122)

< 0.001

1.710 (1.047-2.793)

0.032

Multivariate analysis Model 2

Coexpression pattern of n-FAM83H and n-SCRIB

+/+ (vs. -/-, -/+, +/-)

34/222

2.847 (1.537-5.272)

< 0.001

Abbreviations: CSS, cancer-specific survival; RFS, relapse-free survival; HR, hazard ratio; 95% CI, 95% confidence interval, n-FAM83H; nuclear expression of FAM83H, c-SCRIB; cytoplasmic expression of SCRIB. The variables considered in the multivariate analysis model 1 were preoperative serum level of CEA and CA19-9, tumor stage, T category of tumor stage, lymph node metastasis, distant metastasis, and co-expression pattern of n-FAM83H and c-SCRIB. The variables considered in the multivariate analysis model 2 were preoperative serum level of CEA and CA19-9, tumor stage, T category of tumor stage, lymph node metastasis, distant metastasis, and co-expression pattern of n-FAM83H and n-SCRIB.

  1. In Figures 4A and 4B, which of the survival analyses do they perform in the two panels?

We thank the reviewer for this comment. In response to the reviewer’s comment and to make clear the explanation for Figure 4, we have revised the figure legend for Figure 4. Below is the revised figure legend.

Figure 4. Survival analysis according to individual expression of nuclear FAM83H (n-FAM83H), cytoplasmic FAM83H (c-FAM83H), nuclear SCRIB (n-SCRIB), cytoplasmic FAM83H (c-SCRIB), and co-expression pattern of nuclear FAM83H and cytoplasmic SCRIB (n-FAM83H/c-SCRIB) in the subgroups of CRC patients who received or did not receive adjuvant chemotherapy. (A) Kaplan-Meier survival curves in the 138 patients who did not received chemotherapy. (B) Kaplan-Meier survival curves in the 84 patients who received adjuvant chemotherapy.

  1. In Figure 4, the authors demonstrated that the expression of FAM83H and SCRIB would have a different prognostic effect on the patients with or without chemotherapy. The authors might need to determine if the high expression of FAM83H and/or SCRIB would be associated with the resistance to chemotherapy.

We thank the reviewer for this comment and agree with the reviewer. Usually, patients with advanced cancer received adjuvant chemotherapy. Consequently, adjuvant chemotherapy after the initial operation might affect the prognostic significance of the expression of FAM83H and SCRIB evaluated in tissue specimens obtained before adjuvant chemotherapy. Therefore, the expression of FAM83H and SCRIB would have a different prognostic effect on the patients with or without chemotherapy. In the data presented in Figure 4, the expression of FAM83H and SCRIB significantly predicted CSS and RFS. In addition, although there is a limitation on prognostic significance for RFS, the expression of FAM83H and SCRIB significantly predicted CSS. As the reviewer points out, this result suggests that the high expression of FAM83H and/or SCRIB would be associated with the resistance to chemotherapy. For this point, we agree with the reviewer and we discussed in the discussion section in conjunction with the role of FAM83H and SCRIB for EMT. In addition, in response to the reviewer’s comment, we have inserted the below sentence in the discussion section.

Discussion

With regards to the prognostic significance of FAM83H and SCRIB expression in CRC patients, an interesting finding in this study is that the co-expression pattern of n-FAM83H and c-SCRIB was associated with CSS and RFS in the patients who received adjuvant chemotherapy. This result suggests that higher expression of FAM83H and/or SCRIB would be associated with the resistance to chemotherapy.

  1. The authors might need to explain in more detail why they chose n-FAM83H and c-SCRIB. Based on Figure 3A, the performance of n-FAM83H / c-SCRIB and n-FAM83H / n-SCRIB should be similar.    

We thank the reviewer for this comment. As we respond to the previous question of the reviewer (question 5), the co-expression pattern of n-FAM83H/c-SCRIB had the largest area under the curve among the four types of combined expression patterns. Therefore, based on the result of the ROC analysis, we have chosen the co-expression pattern of n-FAM83H/c-SCRIB. Furthermore, in response to the reviewer’s comment, we have presented the data of the co-expression pattern of n-FAM83H+ and n-SCRIB+ in the results section and Table 5.

Round 2

Reviewer 3 Report

The authors have already addressed all my concerns.